# Pre-exposure prophylaxis uptake concerns in the Democratic Republic of the Congo: Key population and healthcare workers perspectives

Yanhan Shen[1,2]*, Julie Franks[3], William Reidy[3,4], Halli Olsen[3], Chunhui Wang[3], Nadine Mushimbele[5], Richted Tenda Mazala[5], Tania Tchissambou[5], Faustin Malele[5], Apolinaire Kilundu[6], Trista Bingham[7], Gaston Djomand[7], Elie Mukinda[8], Raimi Ewetola[8], Elaine J. Abrams[3,4,9], Chloe A. Teasdale[1,2,4]

1 Department of Epidemiology and Biostatistics, CUNY Graduate School of Public Health and Health Policy, New York, NY, United States of America, 2 Institute for Implementation Science in Population Health, CUNY SPH, New York, NY, United States of America, 3 ICAP at Columbia University, Mailman School of Public Health, New York, NY, United States of America, 4 Department of Epidemiology, Columbia University, Mailman School of Public Health, New York, NY, United States of America, 5 ICAP in Democratic Republic of the Congo, Kinshasa, DRC, 6 Programme National de Lutte contre le VIH/SIDA (PNLS), Kinshasa, DRC, 7 Division of Global HIV & TB, Center for Global Health, Centers for Disease Control and Prevention, Atlanta, GA, United States of America, 8 Democratic Republic of the Congo Centers for Disease Control and Prevention, Kinshasa, DRC, 9 Department of Pediatrics, Columbia University Irving Medical Center, New York, NY, United States of America

* jenny.shen@sph.cuny.edu

## Abstract

Key populations (KP) in the Democratic Republic of the Congo (DRC), including female sex workers (SW), are disproportionally affected by HIV. Quantitative feedback surveys were conducted at seven health facilities in DRC with 70 KP clients enrolled in pre-exposure prophylaxis (PrEP) services to measure benefits and concerns. The surveys also assessed satisfaction with PrEP services and experiences of stigma at the health facilities. Thirty healthcare workers (HCW) were surveyed to measure attitudes, beliefs, and acceptability of providing services to KP. KP client survey participants were primarily female SW. KP clients reported that the primary concern about taking PrEP was fear of side effects (67%) although few KP reported having experienced side effect (14%). HCW concurred with clients that experienced and anticipated side effects were a primary PrEP uptake concern, along with costs of clinic visits.

## Introduction

Key populations (KP), including sex workers (SW), men who have sex with men (MSM), people who inject drugs (PWID) and transgender (TG) women, are disproportionately affected by HIV [1]. In 2019, KP and their sexual partners were estimated to have accounted for 65% of new adult infections worldwide since they are often difficult to reach for critical testing and

**Data Availability Statement:** All relevant data are within the manuscript and its Supporting Information files.

**Funding:** The study was funded by the President's Emergency Plan for AIDS Relief (PEPFAR) through the Centers for Disease Control and Prevention (CDC) under the terms of Cooperative Agreement Numbers U2GGH000994. There was no additional external funding received for this study.

**Competing interests:** The authors have declared that no competing interests exist.

care and treatments services [1]. Risk behaviors such as having multiple sex partners, engaging in unprotected sex, being subjected to forced sex and use of drugs and alcohol contribute to the increased risk of HIV acquisition in these populations [2]. For KP who are HIV-negative, expanding access to and uptake of pre-exposure prophylaxis (PrEP) is a key HIV prevention strategy. PrEP can greatly reduce the incidence of HIV infection [3–7], however expanding access to PrEP in many resource limited settings (RLS) remains challenging [8]. While supply and cost issues are important factors [6,8–12], hesitancy among healthcare workers (HCW) and clients may also be a barrier to PrEP access and uptake [13,14].

In 2020, the Democratic Republic of the Congo (DRC) had an estimated HIV prevalence of 0.7% (95% CI: 0.6%-0.9%) among adults aged 15 to 49 years [15]. KP in DRC are disproportionately affected by HIV [15]. In 2020, HIV prevalence among SW in DRC was 7.5%, which was nine times higher than in the general population [15–17]. In 2019, HIV prevalence among MSM and PWID was estimated at 7.1% and 3.9% [18], respectively, and data from 2016 showed 7.9% HIV prevalence in the TG population in DRC [19]. While PrEP has been approved for use the United States and other countries since 2012 [20,21], PrEP did not become available in DRC until 2018 (the data reported in this paper come from the first pilot or PrEP services in DRC) [22]. In this study, we report descriptive data from surveys conducted with some of the first KP clients who accessed PrEP services at health facilities in DRC to measure perceived benefits and concerns about PrEP. We also present descriptive data from a survey measuring attitudes and acceptability of providing HIV services to KP among healthcare workers (HCW) in the first health facilities offering PrEP in DRC.

## Methods

In collaboration with the DRC Ministry of Health and the Centers for Disease Control and Prevention (CDC), ICAP at Columbia University conducted the first pilot of PrEP services in DRC in 2018. Results on PrEP uptake from a study have been previously reported [22]. The project was conducted between February and November of 2018 at seven 'KP-friendly' health facilities in Kinshasa (four facilities) and Lubumbashi (three facilities) that received support from the CDC through the U.S. President's Emergency Plan for AIDS Relief (PEPFAR) to provide health services for KP clients. To support the implementation of PrEP, health facilities used a comprehensive training package developed for the pilot, which includes a four-day training curriculum for HCW and clinic staff, monitoring and evaluation tools for clinic-level and national reporting of PrEP services, standard operating procedures (SOPs), and job aids for HCW [23]. The 'Pre-exposure Prophylaxis (PrEP) Package' used for the project was developed by ICAP at Columbia University to support the implementation of PrEP within healthcare settings [23]. The training curriculum and tools designed for clinical care providers covers PrEP basics, screening for risk and eligibility, how to conduct initial and follow-up visits, monitoring and managing side effects and HIV seroconversion, recordkeeping and patient tracking, as well as how to reduce PrEP-related stigma. In addition, HCW also received the CDC's HCW sensitization training to increase their understanding of the unique medical and psychosocial needs of KP clients and were trained to use the PEPFAR *Monitoring, Evaluation, and Reporting (MER) 2.0 KP Classification Tool* to help identify KP clients eligible for PrEP [24].

During the pilot project, quantitative feedback surveys were conducted at the seven project health facilities to measure satisfaction and to identify concerns about PrEP services among KP respondents ≥18 years who were able to understand French, Lingala or Swahili. Ten KP clients who had initiated PrEP were sampled per facility using convenience sampling, including clients currently taking PrEP and those who had discontinued. Details of PrEP initiation

are described elsewhere [22]. The survey was interviewer-administered and collected demographic information, including sex at birth, self-identified gender, age, and KP group identification based on the PEPFAR *MER 2.0 KP Classification Tool*, as well as risk behaviors and HIV testing history. Self-identified gender was assessed by asking KP clients "Do you consider yourself to be: a man, woman, transgender or other person?" with options: Man, Woman, Transgender (male to) female, Transgender (female to) male, other, refusal to respond. Current concerns about PrEP use were assessed by collecting respondent's reaction to a 5-item modified Likert scale from published instruments which was originally developed to assess individual level factors associated with adherence to antiretroviral treatment for HIV infection [25–28], adapted to PrEP use [29,30]. Participants reported current PrEP intake, number of days with missed PrEP doses in past seven days, and the reasons for missing doses selected from a list of items. Data were all self-reported including HIV status (medical records were not used to verify information). All participation was voluntary and participants provided verbal consent. Survey data were collected from June to September of 2018 and participants received the equivalent of $10 USD for completing the survey.

In addition to the client survey, HCW at the seven project facilities were also invited to participate in a survey about PrEP services for KP. HCW ≥ 18 years, who were French speaking and had at least 3 months of experience providing HIV-related services to KP at the project facilities were eligible. Convenience samples of 5 HCW per facility were recruited. The HCW survey was adapted from the Health Policy Initiative tool to assess HIV-related stigma and discrimination in health facilities and providers [31] and collected information on participant age and sex. HCW perceptions about clients' PrEP concerns and HCW acceptability of providing services to KP were assessed by a 5-item modified Likert scale ("strongly agree", "agree", "don't know", "disagree", "strongly disagree"). HCW opinions towards "what can be done to improve the service provided to KP in health facilities in the DRC" and "what types of training would you recommend for health professionals" were collected (select all that apply). Attitudes about recommending PrEP were collected by asking "Would you recommend PrEP to a patient, friend, or family member?" with response options "Definitely", "Probably", "Maybe", "Probably not", "Don't know". HCW surveys were self-administered on electronic tablets after instructions were given by the study team. HCW participation was voluntary and verbal consent was taken. No compensation was given to HCW and their personally identifiable information was not recorded.

Descriptive data from the surveys are reported. The project was not designed to present results by KP groups, as such, results are presented using descriptive statistics without tests of statistical significance. The study protocol was approved by the DRC Ministry of Public Health's National Ethics Committee for Health, the Columbia University Irving Medical Center Institutional Review Board. This project was reviewed in accordance with CDC human research protection procedures and was determined to be research, but CDC investigators did not interact with human subjects or have access to identifiable data or specimens for research purposes.

## Results

Seventy participants completed the KP PrEP survey; median age was 31 years [IQR: 28–38] and 58 (83%) self-identified as female gender (**Table 1**). Almost all (96%) PrEP survey respondents reported sale of sex as a main source of income, 9 (13%) reported injection drug use, and 66 (94%) said that they were on PrEP at the time of the survey.

A total of 30 HCW participated in the survey; 20 (67%) were female and most (67%) were 25–34 years of age (**Table 1**). About a third of HCW participants were nurses or midwives and

**Table 1. Characteristics of key population (KP) clients and healthcare workers (HCW) at 7 health facilities providing pre-exposure prophylaxis (PrEP) service in the Democratic Republic of the Congo, 2018.**

| KP Client Survey (N = 70) | | HCW Survey (N = 30) | |
|---|---|---|---|
| Sex at birth, N (%) | | Sex at birth, N (%) | |
| Female | 38 (54%) | Female | 20 (67%) |
| Male | 32 (46%) | Male | 10 (33%) |
| Self-identified gender, N (%) | | Age (years), N (%) | |
| Female | 58 (83%) | < 25 | 1 (3%) |
| Male | 10 (14%) | 25–34 | 20 (67%) |
| Transgender (Male to Female) | 2 (3%) | 35–44 | 5 (17%) |
| | | 45–54 | 4 (13%) |
| KP classification, N (%) | | Professional title as HCW, N (%) | |
| SW | 38 (54%) | Deputy Director | 2 (7%) |
| TG/SW | 14 (20%) | Doctor of Medicine | 6 (20%) |
| TG/SW/PWID | 6 (9%) | Nurse/Midwife | 11 (37%) |
| TG/PWID | 2 (3%) | Worker/Social worker | 5 (17%) |
| SW/PWID | 1 (1%) | Community counselor | 1 (3%) |
| SW/MSM | 8 (11%) | Laboratory Technician | 1 (3%) |
| MSM | 1 (1%) | Other | 2 (7%) |
| | | Refuse to respond | 2 (7%) |
| Age (years), median [IQR] | 31 [28 – 38] | Years working in current position, median [IQR] | 2.5 [2.0–5.0] |
| Selling sex as main income, N (%) | | Years working in current position, N (%) | |
| Yes | 67 (96%) | 1–2 | 14 (47%) |
| No | 3 (4%) | 3–5 | 12 (40%) |
| | | 5–10 | 2 (7%) |
| | | > 10 | 2 (7%) |
| Inject drugs, N (%) | | Years working with HIV+ patients, N (%) | |
| Yes | 9 (13%) | < 1 | 3 (10%) |
| No | 61 (87%) | 1–2 | 12 (40%) |
| | | 3–5 | 12 (40%) |
| | | 5–10 | 1 (3%) |
| | | > 10 | 2 (7%) |
| Currently on PrEP, N (%) | | Years providing HIV services for KP, N (%) | |
| Yes | 66 (94%) | < 1 | 2 (7%) |
| No | 4 (6%) | 1–2 | 15 (50%) |
| | | 3–5 | 12 (40%) |
| | | 5–10 | 0 |
| | | > 10 | 1 (3%) |
| KP PrEP adherence barriers *, N (%) | | | |
| Forgot | 17 (24%) | | |
| Worried about side effects | 12 (17%) | | |
| Had side effects | 10 (14%) | | |
| Ran out of pills | 7 (10%) | | |
| Didn't feel like taking it | 4 (6%) | | |
| Felt the pills were not needed | 4 (6%) | | |
| Other | 8 (11%) | | |
| Refusal to respond | 15 (21%) | | |

Abbreviations: MSM: Men who have sex with men, SW: Sex workers, PWID: People who inject drugs, TG: Transgender.

*Non-exclusive answers.

a quarter were doctors. About half of HCW who participated in the surveys had been working with people living with HIV (PLHIV) and/or KP for less than 2 years.

Among KP clients, almost all (90%) reported that taking PrEP would help them and their partners stay HIV negative and 90% reported that taking PrEP would set a good example for other people (**Fig 1A**). The most frequently reported concern among PrEP clients was possible side effects (67%). In addition, 23% of KP were concerned about experiencing HIV-related stigma and 30% worried about losing social support as a result of taking PrEP. Among HCW,

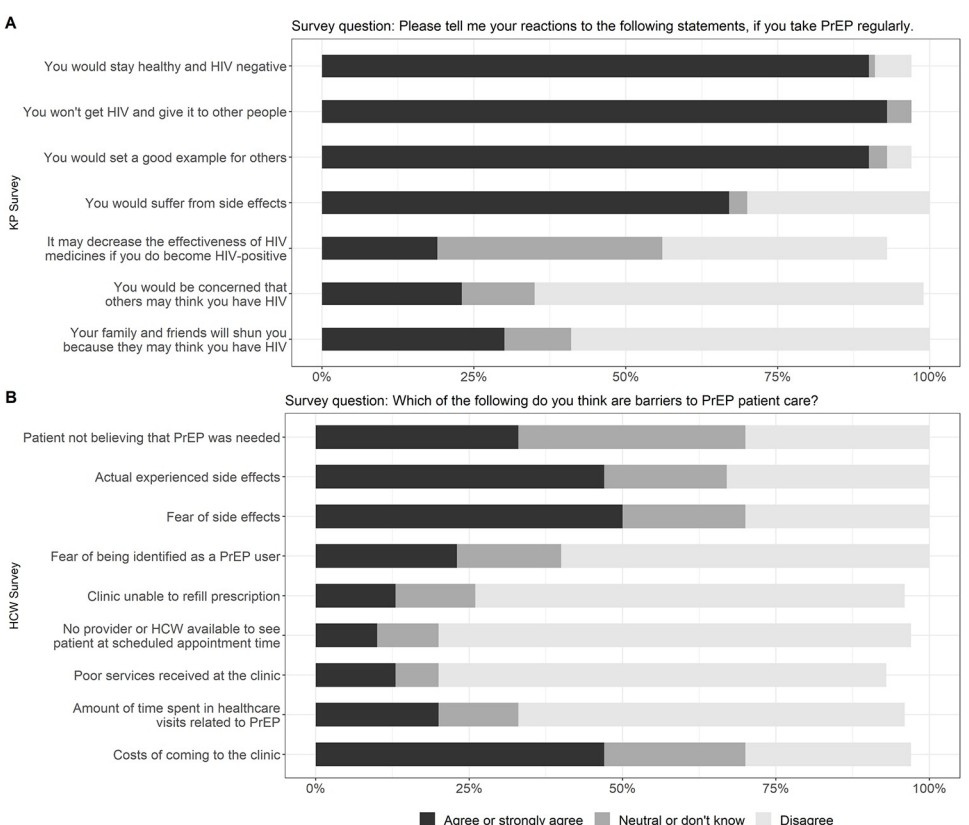

**Fig 1. Key population (KP) clients and healthcare workers (HCW) perceived pre-exposure prophylaxis (PrEP) uptake facilitators and concerns, at 7 health facilities in the Democratic Republic of the Congo, 2018.** Panel A: KP (N = 70) perceived PrEP uptake facilitators and concerns. Panel B: HCW (N = 30) perceived KP PrEP uptake concerns.

actual experiences of PrEP side effects (47%), fear for PrEP side effects (50%), and costs of getting to the clinic (47%) would be barriers to client PrEP uptake (**Fig 1B**).

Reasons for missed PrEP doses over the past 7 days were reported by 55 (79%) of the KP survey respondents (**Table 1**). The most common reasons for missed doses were forgetting (24%), worrying about side effects (17%), and actually experiencing side effects (14%). In addition, KP also reported running out of pills (10%), not wanting to take (6%), and feeling that PrEP was not needed (6%) as reasons for missing PrEP doses.

All (100%) HCW agreed or strongly agreed that they felt comfortable providing care to SW, TG and MSM, whereas 3 (10%) HCW expressed a neutral attitude and 1 (3%) refused to answer towards providing services to PWID (**Fig 2**). Three HCW (10%) reported that KP do not deserve the same quality of healthcare as other patients, and 7 (23%) believed that HIV is a punishment for inappropriate behavior on the part of KP. While the majority (80%) of HCW felt that they were adequately trained to provide high-quality and appropriate care for KP (**Fig 2**), 19 (63%) reported that additional training would help improve services for KP in healthcare facilities (**S1 Table**). Of note, 23 (77%) and 20 (67%) HCW agreed that clinical competency in providing care for KP and communication with KP could be improved, respectively (**S1 Table**). Half of HCW said they would definitely recommend PrEP to a patient, and/or friend, and/or family member (**Fig 3**).

Finally, survey questions about stigma experienced by KP clients showed that 2 (3%) reported having been verbally insulted or harassed within the health facilities, 3 (4%) receiving

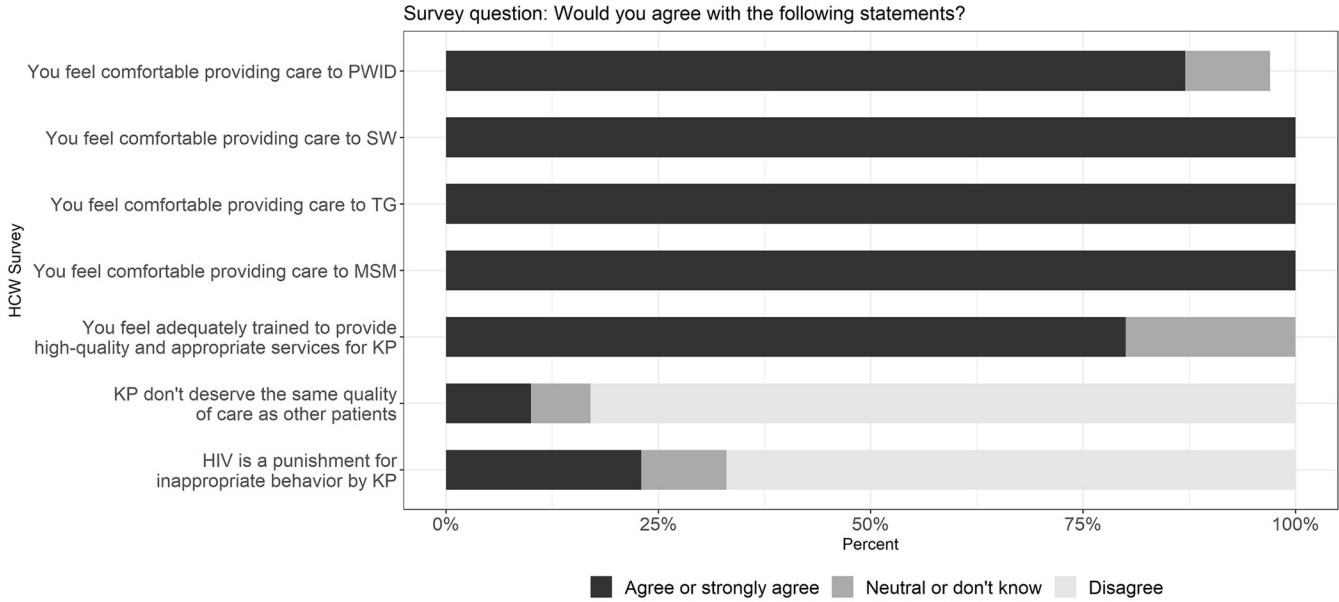

**Fig 2. Healthcare workers (HCW) attitudes towards providing services to key population (KP) clients, in the Democratic Republic of the Congo, 2018.**

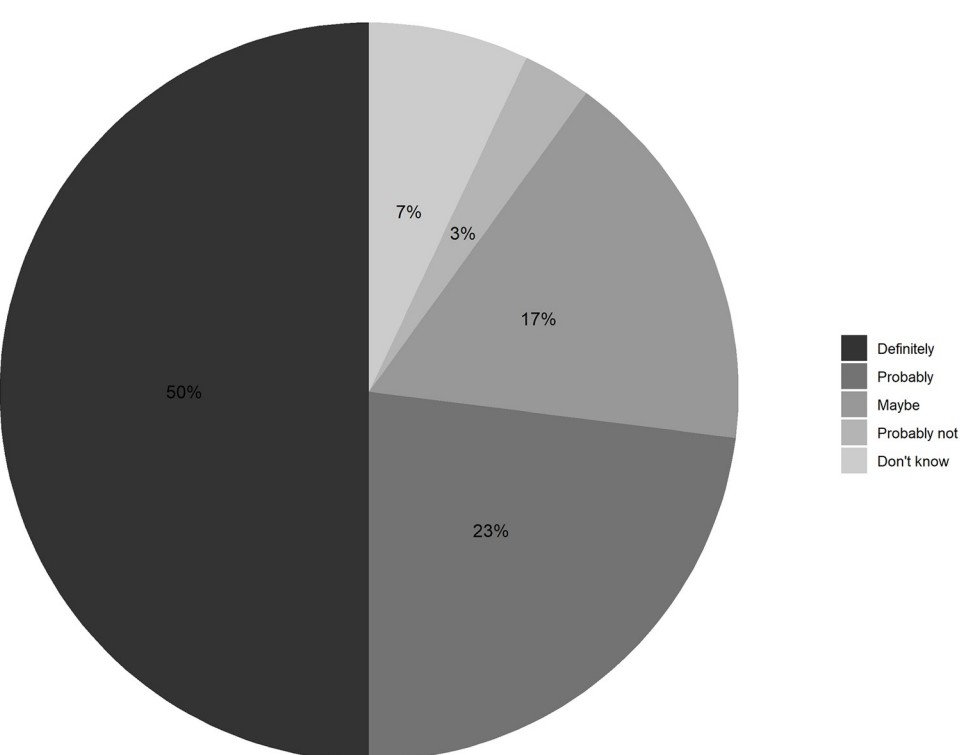

**Fig 3. Healthcare workers (HCW) attitudes towards recommending pre-exposure prophylaxis (PrEP) to a patient, and/or friend, and/or family member, in the Democratic Republic of the Congo, 2018.**

substandard care, and 1 (1%) experiencing lack of confidentiality during their visit on the day of survey. No KP clients reported being treated disrespectfully by HCW.

## Discussion

In this study of KP, primarily female SW accessing the first available PrEP services in the DRC, most agreed that taking PrEP would protect them and their partners from HIV. In addition, while few KP reported actual side effects as a reason for their own lack of PrEP adherence, many reported fear of side effects and concerns about stigma from family and friends as barriers to taking PrEP. Forgetting to take medication was the most commonly reported adherence barrier reported by a third of study respondents taking PrEP. HCW who provide KP health services, including PrEP, were also concerned that perceived and actual experience of side effects would be barriers to client PrEP uptake and adherence. Overall, HCW expressed positive attitudes about PrEP and providing healthcare services to KP.

Similar findings on reasons for and barriers to PrEP uptake among people who could benefit from PrEP have been reported from studies conducted in other African countries. In a qualitative study of female SW and sero-discordant couples in Zimbabwe, perceived HIV risk and concern about acquiring infection were key drivers of PrEP uptake while reasons for declining PrEP included fears of pill burden and side effects, as well as discouragement from family members [32]. Qualitative studies in sero-discordant couples in Kenya and female SW in South Africa also found that fear of side effects and stigma were important barriers to PrEP uptake, along with doubts about its effectiveness [33,34]. Our study also found discordance between perceived concerns about the potential for side effects, which were reported by a majority of KP clients (67%), and actual experience of side effects, which were reported by a much smaller proportion (14%) of the same group. It is very interesting that while most of the KP surveyed had not had side effects themselves while taking PrEP, they remained concerned that side effects would impede other KP clients from taking PrEP, which was also reported in a qualitative study conducted in Eswatini [35]. In part, this discordance may be related to the pilot's implementation of PrEP when relatively little real-world data on its safety and tolerability, especially outside of high-income settings, were available [22]. In this early stage of PrEP roll-out, the World Health Organization recommendations and many national guidelines emphasized caution and thorough client education on possible side effects of PrEP [36]. These findings highlight the importance of educating those who could benefit from PrEP about the low risk of side effects and its overall effectiveness in preventing HIV infection, as well as those who perceive HIV risk and concern about acquiring infection as messengers for PrEP uptake.

Our findings also underscore the continuing challenges related to stigma associated with HIV prevention and treatment interventions, which is further compounded for KP who face additional stigma. Although all HCW felt comfortable providing care to KP clients, almost one-quarter agreed or strongly agreed that HIV is a punishment for inappropriate behavior by KP. This finding from KP-friendly health facilities where staff may be more accepting of KP is concerning as stigma and discrimination against KP clients is likely more prevalent across all DRC health facilities. Data from other settings have shown less comfort among HCW for providing services to KP. In South Africa, only 30.2%, 25.2%, and 27.7% of HCW strongly felt comfortable providing health services for SW, PWIUD, and MSM, respectively [37]. Studies from Ghana and South Africa have suggested that stigma-reduction interventions, including training for HCW, can help reduce negative attitudes among HCW towards KP [38,39]. These efforts are critical as client distrust of healthcare providers has been identified as key barriers to PrEP uptake among KP [40,41]. One strategy to improve PrEP uptake and adherence is building trust and improving information sharing between HCW and clients [9,34,42]. In a

study from South Africa they found that female SW's PrEP acceptability would improve by developing clear education and messaging to accurately convey the concept of PrEP [42].

While most HCW at the seven health facilities said they would recommend PrEP, they also expressed concerns about side effects and the belief that KP clients may not understand that PrEP is effective. In a study from Eswatini (formerly Swaziland), HCW reported concerns that PrEP uptake would reduce condom use and cause HIV drug resistance [43]. These findings highlight the critical importance of providing training for HCW on PrEP to improve their knowledge so that they can appropriately counsel clients about the low frequency of side effects and effectiveness for preventing HIV infection. In addition, strengthening virtual or in-person support groups for KP clients on PrEP could facilitate sharing personal experience and stories about PrEP effectiveness [44]. Further studies are also needed to assess whether HCW feelings and opinions about PrEP influence client uptake and adherence and what can be done to improve HCW awareness and attitudes.

A strength of our study was the inclusion of data on the views of HCW who provide services to KP. In our survey, all HCW at the seven health facilities reported feeling they had adequate training and felt comfortable providing services to MSM, SW, and TG, however some expressed discomfort with caring for PWID. The HCW in our study work in health facilities that receive support to provide care for KP and received sensitivity training, which may help explain the high acceptability of caring for KP and PrEP. These data are critical to understanding potential facilitators and barriers to PrEP uptake in DRC and similar settings. Though there is lack of direct evidence that positive HCW's attitudes towards KP will promote PrEP uptake in this population, a study conducted in Kenya showed healthcare providers' ability to provide high-quality empathetic care have been reported as crucial factor for improving antiretroviral therapy adherence among MSM living with HIV [45].

There are some limitations of our study, including the relatively small sample sizes of 70 KP and 30 HCW who only were able to understand French, Lingala or Swahili. Furthermore, the convenience samples of participants may have resulted in a sample willing to participate who may have had better experiences at these facilities. In addition, we measured barriers to PrEP adherence in the past seven days among KP clients at one point in time and do not have information about how long those clients had been on PrEP. There were also 15 (21%) KP clients who declined to report barriers to PrEP adherence, further limiting our sample size. To assess client attitudes regarding PrEP use, the study relied on instruments developed to assess adherence in randomized controlled trials of PrEP efficacy. This may have limited our ability to understand the constellation of factors that influence real-world decisions to uptake and adhere to PrEP as a proven preventive measure against potential HIV acquisition [46,47]. As noted, the study was also conducted in designated KP-friendly health facilities where staff may be more accepting of KP and supportive of PrEP compared with HCW in clinics serving the general public. As such, our findings may not be generalizable to other settings.

Overall, we found positive attitudes about PrEP among KP clients enrolled in PrEP services and among HCW providing these services. Our results also showed persistent concerns about potential side effects and stigma associated with PrEP. Forgetting to take PrEP was the most commonly reported barrier to adherence, which underscores the need for longer-acting PrEP modalities that are on the horizon. Although most HCW at these KP-friendly health facilities reported feeling adequately trained and comfortable providing care to KP clients, there was still evidence of stigma and discrimination towards KP which could be a barrier to engagement in HIV services among KP. Further efforts are needed to ensure that all people can access the full package of HIV prevention and treatment services without stigma or discrimination.

## Supporting information

**S1 Table. Healthcare workers (HCW) self-reported gaps of improving key population pre-exposure prophylaxis (PrEP) service, in the Democratic Republic of the Congo, 2018.**
(DOCX)

**S1 File.**
(PDF)

**S1 Data.**
(XLSX)

## Acknowledgments

The authors gratefully acknowledge the collaboration of staff members at the KP-friendly clinics in Kinshasa and Lubumbashi whose dedication made this demonstration project possible, as well as the participation of the clinic clients.

## Author Contributions

**Conceptualization:** Julie Franks, William Reidy, Elaine J. Abrams, Chloe A. Teasdale.

**Data curation:** Nadine Mushimbele, Richted Tenda Mazala, Tania Tchissambou, Faustin Malele, Apolinaire Kilundu.

**Formal analysis:** Yanhan Shen, William Reidy, Chunhui Wang, Elaine J. Abrams, Chloe A. Teasdale.

**Funding acquisition:** Julie Franks, William Reidy, Trista Bingham, Gaston Djomand, Elie Mukinda, Raimi Ewetola, Elaine J. Abrams, Chloe A. Teasdale.

**Investigation:** Julie Franks, William Reidy, Halli Olsen, Apolinaire Kilundu, Trista Bingham, Gaston Djomand, Elie Mukinda, Raimi Ewetola, Elaine J. Abrams, Chloe A. Teasdale.

**Methodology:** Julie Franks, William Reidy, Halli Olsen, Chunhui Wang, Trista Bingham, Gaston Djomand, Elie Mukinda, Raimi Ewetola, Chloe A. Teasdale.

**Project administration:** Julie Franks, William Reidy, Halli Olsen, Nadine Mushimbele, Richted Tenda Mazala, Tania Tchissambou, Faustin Malele.

**Resources:** Apolinaire Kilundu, Trista Bingham, Gaston Djomand, Elie Mukinda, Raimi Ewetola.

**Supervision:** Julie Franks, William Reidy, Richted Tenda Mazala, Tania Tchissambou, Faustin Malele, Elaine J. Abrams.

**Writing – original draft:** Yanhan Shen.

**Writing – review & editing:** Julie Franks, William Reidy, Halli Olsen, Chunhui Wang, Nadine Mushimbele, Richted Tenda Mazala, Tania Tchissambou, Faustin Malele, Apolinaire Kilundu, Trista Bingham, Gaston Djomand, Elie Mukinda, Raimi Ewetola, Elaine J. Abrams, Chloe A. Teasdale.

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
