## [Decision Letter · Decision Letter 0]

4 Apr 2023

PONE-D-23-00937Pre-exposure Prophylaxis Uptake Concerns in the Democratic Republic of the Congo: Key Population and Healthcare Workers PerspectivesPLOS ONE

Dear Dr. Shen,

Thank you for submitting your manuscript to PLOS ONE. After careful consideration, we feel that it has merit but does not fully meet PLOS ONE’s publication criteria as it currently stands. Therefore, we invite you to submit a revised version of the manuscript that addresses the points raised during the review process. There are important methodological concerns and inconsistencies in this manuscript. Kindly respond to the reviewers' comments in the revised submission.

We look forward to receiving your revised manuscript.

Kind regards,

Edward Nicol, PhD

Academic Editor

PLOS ONE

Journal Requirements:

"This work was supported by the President’s Emergency Plan for AIDS Relief (PEPFAR) through the Centers for Disease Control and Prevention (CDC) under the terms of Cooperative Agreement Numbers U2GGH000994. JF, WR, HO, CW, NM, RTM, TT, FM, EAJ and CAT received funding for this work. "

Reviewers' comments:

Reviewer's Responses to Questions

**Comments to the Author**

1. Is the manuscript technically sound, and do the data support the conclusions?

Reviewer #1: Partly

Reviewer #2: Yes

2. Has the statistical analysis been performed appropriately and rigorously? 

Reviewer #1: Yes

Reviewer #2: Yes

3. Have the authors made all data underlying the findings in their manuscript fully available?

Reviewer #1: No

Reviewer #2: No

4. Is the manuscript presented in an intelligible fashion and written in standard English?

Reviewer #1: Yes

Reviewer #2: Yes

5. Review Comments to the Author

Reviewer #1: This is a well-written paper that explores Pre-exposure prophylaxis attitudes and concerns among members of key impacted populations for HIV as well as a sample of healthcare workers that provides clinical care and services to key populations in the Democratic Republic of the Congo. The manuscript reports some of the first data from DRC regarding PrEP attitudes, as PrEP services have only recently become available, therefore there is good novelty in presenting emerging data from this region. The authors seek to examine timely and important questions around PrEP attitudes that can inform optimized care, which is of high significance. The exploratory analysis conducted does shed new light on a number of points, including patient specific and HCW barriers to PrEP utilization.

Despite the merits of the paper, there are also some weaknesses that warrant attention.

The introduction of the paper notes HIV prevalence data on FSWs in the DRC, but does not provide background information on other KPs. Are these data available? It would seem particularly important if data on TGW are available since this group is heavily represented in the survey responders for the current study.

In the methods, the authors note that all data was self-reported, no medical records data were utilized, yet the data statement contradicts this. Some clarification appears warranted.

I had some concern regarding the survey instruments that were used to assess attitudes and concerns, these adapted items are from somewhat dated HIV treatment research using the IMB model, and no information was provided on their reliability or validity. Were PrEP instruments actually considered for the data collection? Justification for the choice of measures would strengthen the paper.

In the results section, it was somewhat surprising to see that 67% the KP reported the primary concern of side effects from PrEP, especially since this is a sample of PrEP users. Since these are PrEP users I wondered if this is reflective of actual experienced side effects, and if so, this is quite elevated compared to other reports. The reported adherence barriers seem to indicate that 31% had concerns or actual experienced side effects with the PrEP medication, so this would seem to warrant clarification for the reader. Also, there was a fairly high refusal rate on the adherence item. How should we interpret that? I had some challenges interpreting the items in supplementary Table 1 as well, for example, what is “sensitization of KP”? A few of these items would benefit from further explanation.

The discussion is well written and nicely touches on the role of stigma in PrEP uptake, but in some ways it is fairly general regarding the role of additional training for HCWs. Perhaps the authors could include and reinforce more specific actions needed based on their findings.

Reviewer #2: This manuscript presents descriptive data about the access and use of oral PrEP by key populations in the Democratic Republic of the Congo, from the perspective of PrEP clients and health care workers. With some additional clarifications, this will be an important contribution to the evidence base on real-world implementation of oral PrEP services. Below please find a summary of recommended additions or revisions.

Introduction:

- The introduction is currently very brief and would benefit from additional contextual information. I would recommend adding a paragraph to summarize available data on the main reasons why key populations are disproportionately affected by HIV; a paragraph on any available data on PrEP use among key populations in the DRC; and a paragraph summarizing available information on relevant norms and attitudes surrounding key populations and HIV in the DRC.

- The second paragraph would benefit from the addition of adding data on HIV incidence or prevalence among key populations groups not already mentioned in this paragraph, such as trans individuals, people who use injection drugs, and men who have sex with men. If these data are not available, either a statement about this knowledge gap or a summary of available data on the regional level might be appropriate.

- Please add the aim of this study to the introduction section.

Methods:

-If possible, please link to or describe in further detail the training materials used to prepare providers for PrEP provision. Describing if/how the training addressed stigma, if it included values clarification, and what the KP Classification Tool consisted of would help the reader understand the context in which this research was conducted.

- Please clarify how PrEP clients were asked to report their gender. Were participants given the response options shown in Table 1? This would help the reader understand how the two trans women were identified.

- Table 1 states that the study was conducted in 7 health facilities. Is it possible to provide further details about these facilities? For example, are they KP-focused facilities, are they located in an urban or rural setting, etc.

Results:

-The part of Table 1 that shows KP classification would be better represented visually, if possible. I would encourage the authors to consider creating a diagram of overlapping circles to indicate the multiple and overlapping KP identities.

- Regarding client barriers to PrEP adherence, there is a discrepancy in how data are reported in Table 1 and how they are reported in the body of the manuscript. It appears as though a different denominator is used. It would be more straightforward for the reader if these percentages were the same in these two places.

Discussion:

- There were some findings that I personally found interesting that were not discussed at length in the discussion. I would encourage the authors to consider discussing the following finding: Although all providers were comfortable providing PrEP to most KP groups, almost one-quarter of providers believed that HIV is a punishment for inappropriate behavior. This seems to further emphasize the need to improve provider attitudes as discussed in the third paragraph of the discussion.

- The final paragraph of the discussion points to PrEP-associated stigma. It is worth discussing whether the authors expect there to be any KP-specific stigma also at play.

6. PLOS authors have the option to publish the peer review history of their article (what does this mean?). If published, this will include your full peer review and any attached files.

Reviewer #1: No

Reviewer #2: **Yes: **Kathleen Ridgeway

---

## [Author Response · Author response to Decision Letter 0]

19 May 2023

Responses to the reviewer comments

Submission related requirements

Response: We have formatted the manuscript and file names accordingly. 

"This work was supported by the President’s Emergency Plan for AIDS Relief (PEPFAR) through the Centers for Disease Control and Prevention (CDC) under the terms of Cooperative Agreement Numbers U2GGH000994. JF, WR, HO, CW, NM, RTM, TT, FM, EAJ and CAT received funding for this work. "

Response: Please see our revised funding statement in the cover letter. 

Response: In our resubmission, we have provided an Excel file that contains the study data (in two spreadsheets) and a PDF file with both survey questionnaires. Thank you for your assistance in making the data available. 

Response: We have included the data in a newly added figure (Figure 3). 

Response: We have included caption(s) for supporting information after the DISCUSSION section and updated the in-text citation accordingly. 

Reviewers’ Comments to the Authors

Reviewer #1: 

1. The introduction of the paper notes HIV prevalence data on FSWs in the DRC, but does not provide background information on other KPs. Are these data available? It would seem particularly important if data on TGW are available since this group is heavily represented in the survey responders for the current study.

Response: Thank you for your input. We have added the following information to the Introduction (paragraph 2): “KP in DRC are disproportionately affected by HIV [15]. In 2020, HIV prevalence among SW in DRC was 7.5%, which was nine times higher than in the general population [15–17]. In 2019, HIV prevalence among MSM and PWID was estimated at 7.1% and 3.9% [18], respectively, and data from 2016 showed 7.9% HIV prevalence in the TG population in DRC [19].”

2. In the methods, the authors note that all data was self-reported, no medical records data were utilized, yet the data statement contradicts this. Some clarification appears warranted.

Response: We apologize this is an error. All data were self-reported (no data from medical charts were used). We have corrected the data statement in the cover letter and have provided the survey data. 

3. I had some concern regarding the survey instruments that were used to assess attitudes and concerns, these adapted items are from somewhat dated HIV treatment research using the IMB model, and no information was provided on their reliability or validity. Were PrEP instruments actually considered for the data collection? Justification for the choice of measures would strengthen the paper.

Response: Thank you for this pertinent observation. We have clarified the source of the instrument in the text and citations. While the instrument is derived from IMB constructs to explain ART adherence, it was adapted by IMB experts to assess motivations and barriers to PrEP use in global RCTs of PrEP efficacy. At the time the protocol for our study was developed, there were few published data on factors associated with use of PrEP as a demonstrated effective HIV prevention tool in real-world contexts. Research to understand PrEP uptake has advanced rapidly since this study was conducted. We have referenced reliance on ART-focused instruments as a potential limitation of the study, and cited recent work to improve our understanding of PrEP use.

We revised the following sentence in METHOD paragraph 2: 

“Current concerns about PrEP use were assessed by collecting respondent’s reaction to a 5-item modified Likert scale from published instruments which was originally developed to assess individual level factors associated with adherence to antiretroviral treatment for HIV infection [25–28], adapted to PrEP use [29,30]”

We added the following sentence in DISCUSSION limitation paragraph:

“To assess client attitudes regarding PrEP use, the study relied on instruments developed to assess adherence in randomized controlled trials of PrEP efficacy. This may have limited our ability to understand the constellation of factors that influence real-world decisions to uptake and adhere to PrEP as a proven preventive measure against potential HIV acquisition [46,47].”

4. In the results section, it was somewhat surprising to see that 67% the KP reported the primary concern of side effects from PrEP, especially since this is a sample of PrEP users. Since these are PrEP users I wondered if this is reflective of actual experienced side effects, and if so, this is quite elevated compared to other reports. The reported adherence barriers seem to indicate that 31% had concerns or actual experienced side effects with the PrEP medication, so this would seem to warrant clarification for the reader. 

Response: We agree that these findings (which were checked to ensure accuracy) warrant further comment.. We have added a note in the discussion: “Our study also found discordance between perceived concerns about the potential for side effects, which were reported by a majority of KP clients (67%), and actual experience of side effects, which were reported by a much smaller proportion (31%) of the same group. It is very interesting that while most of the KP surveyed had not had side effects themselves while taking PrEP, they remained concerned that side effects would impede other KP clients from taking PrEP, which was also reported in a qualitative study conducted in Eswatini [35]. In part, this discordance may be related to the pilot’s implementation of PrEP when relatively little real-world data on its safety and tolerability, especially outside of high-income settings, were available [22]. In this early stage of PrEP roll-out, the World Health Organization recommendations and many national guidelines emphasized caution and thorough client education on possible side effects of PrEP [36]..”

5. Also, there was a fairly high refusal rate on the adherence item. How should we interpret that? 

Response: Thank you for noting this. We have added the following sentence in the Discussion section limitation paragraph. “There were also 15 (21%) KP clients who declined to report barriers to PrEP adherence, further limiting our sample size..”

6. I had some challenges interpreting the items in supplementary Table 1 as well, for example, what is “sensitization of KP”? A few of these items would benefit from further explanation.

Response: Thank you for pointing this out. We have added a footnote in supplementary Table 1 (S1 Table) to explain the implication of “additional sensitization on KP”. 

7. The discussion is well written and nicely touches on the role of stigma in PrEP uptake, but in some ways it is fairly general regarding the role of additional training for HCWs. Perhaps the authors could include and reinforce more specific actions needed based on their findings.

Response: Thank you for the suggestions. We have revised the Discussion paragraph 4 to reinforce the actions based on our findings: 

“These findings highlight the critical importance of providing training for HCW on PrEP to improve their knowledge so that they can appropriately counsel clients about the low frequency of side effects from PrEP and its effectiveness for preventing HIV infection. In addition, strengthening virtual or in-person support groups for KP clients on PrEP could facilitate sharing personal experience and stories about PrEP effectiveness [44]. Further studies are also needed to assess whether HCW feelings and opinions about PrEP influence client uptake and adherence and what can be done to improve HCW awareness and attitudes.”

Reviewer #2: 

1. Introduction: The introduction is currently very brief and would benefit from additional contextual information. I would recommend adding a paragraph to summarize available data on the main reasons why key populations are disproportionately affected by HIV; a paragraph on any available data on PrEP use among key populations in the DRC; and a paragraph summarizing available information on relevant norms and attitudes surrounding key populations and HIV in the DRC.

Response: Thank you for your input. We have additional information about these issues in the revised manuscript.

In Introduction paragraph 1, we have added the following sentences to describe the main reasons why kep populations are disproportionately affected by HIV: 

“Risk behaviors such as having multiple sex partners, engaging in unprotected sex, being subjected to forced sex and use of drugs and alcohol also contribute to the increased risk of HIV acquisition in these populations.”

We have also added the following to the Introduction paragraph 2 to describe the data on PrEP use among KP in DRC:

“KP in DRC are disproportionately affected by HIV [15]. In 2020, HIV prevalence among SW in DRC was 7.5%, which was nine times higher than in the general population [15–17]. In 2019, HIV prevalence among MSM and PWID was estimated at 7.1% and 3.9% [18], respectively, and data from 2016 showed 7.9% HIV prevalence in the TG population in DRC [19]. While PrEP has been approved for use the United States and other countries since 2012 [20,21], PrEP did not become available in DRC until 2018 (the data reported in this paper come from the first pilot or PrEP services in DRC) [22].”

2. The second paragraph would benefit from the addition of adding data on HIV incidence or prevalence among key populations groups not already mentioned in this paragraph, such as trans individuals, people who use injection drugs, and men who have sex with men. If these data are not available, either a statement about this knowledge gap or a summary of available data on the regional level might be appropriate.

Response: Thank you for your input. We have added the information noted above to the introduction on HIV prevalence among specific KPs groups with available data. 

3. Please add the aim of this study to the introduction section.

Response: Thank you for the suggestion. We have revised the last sentence of the Introduction section to clarify the aims, 

“In this study, we report descriptive data from surveys conducted with some of the first KP clients who accessed PrEP services at health facilities in DRC to measure perceived benefits and concerns about PrEP. We also present descriptive data from a survey measuring attitudes and acceptability of providing HIV services to KP among healthcare workers (HCW) in the first health facilities offering PrEP in DRC.”

4. Methods:

-If possible, please link to or describe in further detail the training materials used to prepare providers for PrEP provision. Describing if/how the training addressed stigma, if it included values clarification, and what the KP Classification Tool consisted of would help the reader understand the context in which this research was conducted.

Response: Thank you for this suggestion. We have added additional information about the HCW training and have added a citation with the link to the ICAP PrEP training package. 

Additional training description (Methods, first paragraph): 

“The ‘Pre-exposure Prophylaxis (PrEP) Package’ used for the project was developed by ICAP at Columbia University to support the implementation of PrEP within healthcare settings. The training curriculum and tools designed for clinical care providers cover PrEP basics, screening for risk and eligibility, how to conduct initial and follow-up visits, monitoring and managing side effects and HIV seroconversion, recordkeeping and patient tracking, as well as how to reduce PrEP-related stigma.”

5. Please clarify how PrEP clients were asked to report their gender. Were participants given the response options shown in Table 1? This would help the reader understand how the two trans women were identified.

Reponse: Thank you. We have added the following sentence in method paragraph 2. 

“Self-identified gender was assessed by asking KP clients “Do you consider yourself to be: a man, woman, transgender or other person?” with options: Man, Woman, Transgender (male to) female, Transgender (female to) male, other, refusal to respond.”

6. Table 1 states that the study was conducted in 7 health facilities. Is it possible to provide further details about these facilities? For example, are they KP-focused facilities, are they located in an urban or rural setting, etc.

Response: As the survey reports data from vulnerable populations, we prefer to not include the names of the health facilities in order to protect client and HCW privacy.

7. Results:

-The part of Table 1 that shows KP classification would be better represented visually, if possible. I would encourage the authors to consider creating a diagram of overlapping circles to indicate the multiple and overlapping KP identities.

Response: We appreciate your input. Given the small sample size and that most of the clients were either SW or TG/SW, we prefer to show the data in Table 1 rather than as a figure.

8. Regarding client barriers to PrEP adherence, there is a dipancy in how data are reported in Table 1 and how they are reported in the body of the manuscript. It appears as though a different denominator is used. It would be more straightforward for the reader if these percentages were the same in these two places.

Response: Thank you for noting this. We have revised the text (result paragraph 4) to make sure the denominator is used consistently with Table 1.

9. Discussion:

- There were some findings that I personally found interesting that were not discussed at length in the discussion. I would encourage the authors to consider discussing the following finding: Although all providers were comfortable providing PrEP to most KP groups, almost one-quarter of providers believed that HIV is a punishment for inappropriate behavior. This seems to further emphasize the need to improve provider attitudes as discussed in the third paragraph of the discussion.

Response: We have added the following sentences in Discussion paragraph 3. 

“Although all HCW felt comfortable providing care to KP clients, almost one-quarter agreed or strongly agreed that HIV is a punishment for inappropriate behavior by KP. This finding from KP-friendly health facilities, where staff may be more accepting of KP, is concerning as stigma and discrimination against KP clients is likely more prevalent across all DRC health facilities.”

10. The final paragraph of the discussion points to PrEP-associated stigma. It is worth discussing whether the authors expect there to be any KP-specific stigma also at play.

Response: Thank you for this suggestion. We have revised the last sentence in the Discussion as follows, 

“Although most HCW at these KP-friendly health facilities reported feeling adequately trained and comfortable providing care to KP clients, there was still evidence of stigma and discrimination towards KP which could be a barrier to engagement in HIV services among KP. Further efforts are needed to ensure that all people can access the full package of HIV prevention and treatment services without stigma or discrimination.”

---

## [Decision Letter · Decision Letter 1]

17 Oct 2023

Pre-exposure Prophylaxis Uptake Concerns in the Democratic Republic of the Congo: Key Population and Healthcare Workers Perspectives

PONE-D-23-00937R1

Dear Dr. Shen,

We’re pleased to inform you that your manuscript has been judged scientifically suitable for publication and will be formally accepted for publication once it meets all outstanding technical requirements.

Kind regards,

Edward Nicol, PhD

Academic Editor

PLOS ONE

Additional Editor Comments (optional):

Reviewers' comments:

Reviewer's Responses to Questions

**Comments to the Author**

1. If the authors have adequately addressed your comments raised in a previous round of review and you feel that this manuscript is now acceptable for publication, you may indicate that here to bypass the “Comments to the Author” section, enter your conflict of interest statement in the “Confidential to Editor” section, and submit your "Accept" recommendation.

Reviewer #1: All comments have been addressed

Reviewer #2: All comments have been addressed

2. Is the manuscript technically sound, and do the data support the conclusions?

Reviewer #1: Yes

Reviewer #2: Yes

3. Has the statistical analysis been performed appropriately and rigorously? 

Reviewer #1: Yes

Reviewer #2: N/A

4. Have the authors made all data underlying the findings in their manuscript fully available?

Reviewer #1: Yes

Reviewer #2: Yes

5. Is the manuscript presented in an intelligible fashion and written in standard English?

Reviewer #1: Yes

Reviewer #2: Yes

6. Review Comments to the Author

Reviewer #1: Thank you for the careful attention and response to initial review comments. Two minor notes: Figure 3 appears to be without a title, this should be added. Also, please clarify in the abstract and full manuscript the % who experienced PrEP side effects; it is stated that 31% experienced side effects but Table 1 shows this to be 14%. I believe that 31% includes those who were concerned about side effects even though they did not experience them, this should be made clear for the reader.

Reviewer #2: (No Response)

7. PLOS authors have the option to publish the peer review history of their article (what does this mean?). If published, this will include your full peer review and any attached files.

Reviewer #1: No

Reviewer #2: **Yes: **Kathleen Ridgeway

---

## [Editor Report · Acceptance letter]

23 Oct 2023

PONE-D-23-00937R1 

Pre-exposure Prophylaxis Uptake Concerns in the Democratic Republic of the Congo: Key Population and Healthcare Workers Perspectives 

Dear Dr. Shen:

I'm pleased to inform you that your manuscript has been deemed suitable for publication in PLOS ONE. Congratulations! Your manuscript is now with our production department. 

Kind regards, 

on behalf of

Dr. Edward Nicol 

Academic Editor

PLOS ONE